# Responses and sensitivities of maize phenology to climate change from 1971 to 2020 in Henan Province, China

**Ning Zhang**[1,2]*, **Yizhong Qu**[3], **Zhizhong Song**[4,5]*, **Yahui Chen**[6]*, **Jiang Jiang**[6]

**1** School of Architecture and Civil Engineering, Xiamen University, Xiamen, Fujian, China, **2** Faculty of Forestry, The University of British Columbia, Vancouver, British Columbia, Canada, **3** AIR Worldwide Co, Beijing, China, **4** The Engineering Research Institute of Agriculture and Forestry, Ludong University, Yantai, Shandong, China, **5** Department of Plant Science, University of Cambridge, Cambridge, United Kingdom, **6** Collaborative Innovation Center of Sustainable Forestry in Southern China of Jiangsu Province, Nanjing Forestry University, Nanjing, Jiangsu, China

* zhanging@student.ubc.ca (NZ); 3614@ldu.edu.cn (ZS); chenyahui01@163.com (YC)

**Data Availability Statement:** All relevant data are within the paper and its Supporting Information files.

## Abstract

Climate change affects many aspects of the physiological and biochemical processes of growing maize and ultimately its yield. A comprehensive climate suitability model is proposed that quantifies the effects of temperature, precipitation, solar radiation, and wind in different phenological stages of maize. It is calibrated using weather and yield data from China's Henan Province. The comprehensive suitability model showed the capability of correctly hindcasting observed temporal and spatial changes in maize phenology in response to climatic factors. The predicted yield based on the suitability model can well match the recorded field yield very well from 1971–2020. The results of correlation showed that the yields are more closely related to multi-weather factors, temperature and precipitation than to solar radiation and wind. The sensitivity analysis illustrates that temperature and precipitation are the dominant weather factors affecting yield changes based on a direct differentiation method. The comprehensive suitability model can provide a scientific support and analysis tool for predicting grain production considering climate changes.

## Introduction

Agriculture is an industry especially sensitive to climate change [1, 2]. Since the 1960s, climate change has had a greater impact on agricultural production. The changes in global climate make it important to study the meteorological suitability of each growth period of a crop to guide agricultural production [3, 4]. Nowadays, evaluating the suitability of meteorological conditions has gradually shifted from qualitative to quantitative [5–8]. Some scholars have studied the suitability evaluation models of meteorological conditions based on local climate conditions and established various crop climate suitability, evaluation models [9–12]. According to the simulation and prediction of crop growth period, suitability evaluation models can provide some commands to manage farmland timely, choosing appropriate genotypes, and forecasting yields at each growth stage of crops based on agrometeorological conditions.

**Funding:** The author(s) received no specific funding for this work.

**Competing interests:** The authors have declared that no competing interests exist.

Many scholars have researched the assessment of the climate suitability of crops. The climatic suitability of crops is mainly based on the membership function method in fuzzy mathematics [13–15]. Through the establishment of a series of quantitative indices or mathematical models, the quantitative changes of the main agricultural climatic factors (solar radiation, precipitation, and temperature) are transformed into the effects of crop growth and development, yield, and quality [16, 17]. Gu et al. [18, 19] have proposed the concept of agroclimatic suitability describing a region's suitability for growing particular crops such as maize and wheat. Gonzalez et al. [20] have established a model to simulate the growth of crops and the effects of climatic factors. It nicely represents the impact of precipitation and temperature on the growth period of rice in Spain [20]. Olesen et al. [21] study the effect of extreme climate events on agricultural production and the interaction with the surrounding natural ecosystems. Lamptey [22] uses a water suitability model to explore the effect of precipitation on crop yields and provides several ways to improve the utilization of agricultural water resources and rational use of renewable energy to promote stable agricultural production. Wang et al. [23] used meteorological data spanning 1961 to 2000 to fit a suitability model suitable for the main meteorological factors at Datong in China's Shanxi Province. They proposed three hierarchical models of agroecological climate. Matthews and Wassmann [24] analyzed the temporal evolution of the climate's suitability for rice in the Philippines over the past 100 years and induced a model relating climate change to rice yield. He predicts that the adaptability of Philippine rice to climate change will continue to decline.

Crop yields are generally affected by a variety of climatic factors, especially by temperature, precipitation, solar radiation, and wind speed. A few researchers have proposed a development simulation model based on growing degree-days (GDD) and physiological development time (PDT). The model is mechanistic and takes into account thermal effects, insolation, and different varieties [25–27]. Zhao et al. [28] have used fuzzy mathematics to build a comprehensive climate suitability model for winter wheat in China's Henan Province, establishing temperature, solar radiation, and precipitation suitability functions. Yao et al. [29] analyzed the solar radiation, temperature, and water climate suitability and its temporal and spatial distribution laws in each growth period of Gansu Province's corn and Hebei Province's cotton in the past 40 years according to the requirement for solar radiation, temperature, and precipitation in different growth stages of crops. There has not, however, been systematic research on the climate's suitability for growing maize in Henan.

This paper establishes a comprehensive climate suitability model to quantify the effect of temperature, water, solar radiation, and wind in a maize growth cycle based on the maize physiological growth model. The model is used to predict the effect of climatic factors on the yield of corn in Henan in the past 50 years. This model is combined with the direct differentiation method to analyze the climatic factor sensitivities on the yield of corn. The practical purpose of this research is to be able to provide a tool for accurately predicting corn yield in the face of climate change.

## Materials and methods

### A comprehensive climate suitability (CCS) model for maize

According to the growth characteristics of maize in China's Henan Province, its whole growth period can be divided: seedling stage (conventionally June 1 to June 15), heading stage (June 16 to July 15), pollination stage (July 16 to August 15) and maturity stage (August 16 to September 15). A comprehensive suitability model is proposed in the growth period to study the effect of temperature, precipitation, solar radiation, and wind speed on the yield of maize. In actual crop growth, those four factors are not completely independent. There are certain

correlations between temperature and precipitation, temperature and insolation, temperature and wind speed, and of course precipitation and solar radiation. Therefore, the independent suitability models need to be first established for the four factors.

**Temperature suitability model.** To quantitatively analyze the satisfaction degree of heat resources to the growth of maize, a useful temperature suitability model can be defined as [30],

$$F(t)_i = \frac{(t_i - t_L)(t_H - t_i)^B}{(t_0 - t_L)(t_H - t_0)^B} \tag{1}$$

where

$$B = \frac{(t_H - t_0)}{t_0 - t_L} \tag{2}$$

and $F(t)_i$ is the temperature suitability index on the $i^{th}$ day in the maize growth period, $t_i$ is the daily average temperature recorded by the climate station. The variables $t_L$, $t_H$, $t_0$ are the daily minimum, maximum and optimum temperature for the current growth stage of maize, i.e., the three basic point temperatures, which are determined by the temperature requirement of maize at different growth stages. This paper provides a three basic point temperature of four stages based on the comprehensive reference of several pieces of literature [30], as shown in Table 1.

The annual average temperature suitability index for maize can then be calculated as

$$T = \left( \sum_{i=1}^{n} F(t)_i \right) / n, \tag{3}$$

where $n$ is the number of days in the entire growth cycle of maize—107 days. The index $T$ ranges between 0 and 1, with $T = 1$ indicating that temperature has no negative effect on the growth of maize.

**Precipitation suitability model.** Water plays an important role in many physiological processes of maize. Excessive rain leads to slender maize stems and poor tenacity, while drought will inhibit maize growth, reducing the photosynthetic rate and dry matter accumulation [31–33]. Summer in Henan has high temperatures and high evaporation. The water consumption of the whole growth period is 350–450mm, but it differs over the four growth stages. Less than 70% of that is considered mild drought, and more than 130% is mild waterlogging. Wei et al. [34] suggest that the precipitation suitability of maize can be quantified as

$$F(r)_i = \begin{cases} r_i/r_{0i} & r_i < 0.7r_{0i} \\ 1 & 0.7r_{0i} < r_i < 1.3r_{0i} \\ r_{0i}/r_i & r_i > 1.3r_{0i} \end{cases} \tag{4}$$

where $F(r)_i$ is the precipitation suitability index in the $i^{th}$ maize growth period, $r_i$ is the

**Table 1. The three basic temperatures at each growth stage of maize.**

| Growth stage | Period | $t_L$ (°C) | $t_H$ (°C) | $t_0$ (°C) |
|---|---|---|---|---|
| Seedling stage | June 1—June 15 | 17 | 35 | 25 |
| Heading stage | June 16—July 15 | 21 | 35 | 26 |
| Pollination stage | July 16—Aug 15 | 22 | 35 | 26 |
| Mature stage | August 16—September 15 | 18 | 32 | 22 |

**Table 2. Water requirement of maize at different development stages.**

| Growth stage | Period $s_i$ (day) | Water requirement (mm) |
|---|---|---|
| Seedling stage | 15 | 30 |
| Heading stage | 30 | 100 |
| Pollination stage | 31 | 150 |
| Mature stage | 31 | 150 |

accumulated precipitation in the $i$th growth period, and $r_0$ is the period's water requirement for maize. The water requirements in different growth stages of maize are shown in Table 2.

The annual average precipitation suitability index for maize can be calculated as

$$R = \left( \sum_{i=1}^{4} F(r)_i * s_i \right) / 4 \tag{5}$$

where $s_i$ is the number of days in growing stage $i$. Similarly, the index $R$ then also ranges between 0 and 1, with $R = 1$ indicating that precipitation has no negative effect on maize growth.

**Solar radiation suitability model.** Solar radiation is the energy source for crop growth, as well as for the processes of accumulation, distribution, and transfer of photosynthetic products that determine maize yield [35, 36]. Solar radiation enhanced leaf photosynthesis of maize during the daytime, which lead to more photosynthetic products being carried to roots to promote the growth process of maize [37, 38]. As a C4 plant, the carboxylation pathway of maize photosynthesis has a strong correlation with solar radiation [39]. Generally, the development period of maize is advanced under 8 to 9 hours of daily solar radiation. When the solar radiation is inadequate, the nutritional growth slows down and yield decreases. Therefore, sufficient solar radiation can promote the high yield of maize. The solar radiation hours over the whole growth period average in Henan province are 600–900h. In a certain range, the more solar radiation in the whole growth period, the higher the yield. According to the previous research results, 70% of the theoretical solar radiation is the most suitable [40]. Li et al. [40] suggest the following solar radiation suitability model.

$$F(s)_i = \begin{cases} e^{-[(s_i - s_0)/b]^2} & S_i < S_0 \\ 1 & S_i \geq S_0 \end{cases} \tag{6}$$

where $F(s)_i$ is the solar radiation suitability index of the $i$th day in a maize growth period and $s_i$ is the hours of solar radiation length on that day. The variable $S_0$ is the critical solar radiation representing 70% of the theoretical maximum. $b$ is an empirically-derived coefficient, and its reference values are shown in Table 3 [40]. The index $F(s)_i$ is also a membership function ranging from 0 to 1. When the actual hours of sunshine exceed $S_0$, $F(s)_i$ is 1 and the response of maize to solar radiation is the most suitable.

The annual average solar radiation suitability index of maize can then be calculated as,

$$S = \left( \sum_{i=1}^{n} F(s)_i \right) / n \tag{7}$$

then also ranges between 0 and 1, with $S = 1$ indicating the ideal insolation.

**Wind suitability model.** Wind will affect the yield of maize crops in two ways. 1) During flowering and pollination (usually about August 1-August 15), strong wind (wind speed greater than 10.0 m·s$^{-1}$) is easy to cause maize stigma dryness. 2) During the mature period of maize, strong wind (such as in a typhoon) can easily cause maize lodging [41]. Generally,

Table 3. The critical length of solar radiation ($S_0$) and the fitting coefficient $b$ in the different growth stages of maize in Henan.

| Growth stage | Length of critical solar radiation (h) | Empirical coefficient |
|---|---|---|
| Seedling stage | 8.85 | 4.77 |
| Heading stage | 9.42 | 5.08 |
| Pollination stage | 9.53 | 5.14 |
| Mature stage | 9.71 | 5.24 |

Henan Province is not affected by typhoons, and the problem of maize lodging can be solved by planting a suitable cultivar [42]. In the current study, the focus was made on flowering and pollination from August 1 to August 15. During that period the most suitable wind speed is less than 6.0 m·s⁻¹ [43]. When the wind speed is greater than 10.0 m·s⁻¹, the stigma will dry up, affecting the yield. The wind suitability model for maize can be expressed as [43]

$$F(w)_i = \begin{cases} 1 & w_i < w_L \\ \dfrac{w_i - w_L}{w_H - w_L} & w_L \leq w_i < w_H \\ 0 & w_H \leq w_i \end{cases} \quad (8)$$

where $F(w)_i$ is the wind suitability index on the $i^{th}$ day in a maize growth period, $w_L$ and $w_H$ are the lowest and the highest critical wind speeds. The annual average wind suitability index for maize can then be calculated as

$$W = \left( \sum_{i=1}^{nw} F(w)_i \right) / nw \quad (9)$$

where $nw$ is the number of days in the flowering period—15 days [43].

**A comprehensive reduced yield model.** The ideal temperature, water, solar radiation and wind differ in maize's different growth stages, and the contribution coefficient of each meteorological parameter is also different [9, 43]. In addition, these four climatic factors are not completely independent in the actual crop growth period. Temperature and precipitation, temperature and solar radiation and of course precipitation and solar radiation have a certain correlation. The reduced yield index caused by a single weather factor can be calculated using the appropriate independent climatic suitability model, The temperature reduced yield index $D_t = 1-T$, the precipitation reduced yield index $D_r = 1-R$, the solar radiation reduced yield index $D_s = 1-S$, and wind reduced yield index $D_w = 1-W$.

A comprehensive reduced yield index can be calculated as a weighted average of the four single climatic reduced yield indices,

$$D = b_t D_t + b_r D_r + b_s D_s + b_w D_w + b_{tr} D_t D_r + b_{ts} D_t D_s + b_{rs} D_r D_s \quad (10)$$

where $b_t$, $b_r$, $b_s$, $b_w$, $b_{tr}$, $b_{ts}$ and $b_{rs}$ are the weights of individual or coupled climatic reduced yield indices. They need to be determined empirically by the reverse analysis based on recorded and predicted yields.

## Sensitivities of comprehensive reduced yield model

The climatic factors that affect the maize yield are often diverse [3, 4]. Due to the growth characteristics of maize in four growth stages being quite different and the whole growth cycle is longer, the maize yield is more sensitive to climate change. The sensitivity of any phenological change refers to the days over which the maize's phenology changes as the weather factors by

one unit. In the current study, a direct differentiation method can be used to quantify the sensitivity of maize weather yields and the comprehensive reduced yield index obtained from the comprehensive reduced yield model. The sensitivity of the comprehensive reduced yield index concerning a single weather variable can be calculated as

$$\xi = \frac{\partial D}{\partial \theta} \tag{11}$$

Taking temperature as an example, i.e., $\theta = t$,

$$\xi = \frac{\partial D}{\partial t} = b_t \frac{\partial D_t}{\partial t} + b_{tr} D_r \frac{\partial D_t}{\partial t} + b_{ts} D_s \frac{\partial D_t}{\partial t} \tag{12}$$

where,

$$\frac{\partial D_t}{\partial t} = -\frac{\partial T}{\partial t} = \left( \sum_{i=1}^{n} \frac{\partial F(t)_i}{\partial t} \right) / n \tag{13}$$

and

$$\frac{\partial F(t)_i}{\partial t} = -t_L \frac{(t_H - t_i)^B}{(t_0 - t_L)(t_H - t_0)^B} \tag{14}$$

Similarly, the sensitivities of the comprehensive reduced yield index to other weather variables can be calculated similarly.

## Results and discussion

The comprehensive suitability model can be employed to hindcast maize yields in Henan Province from 1971 to 2020. As shown in Fig 1, the black line represents the maize raw yield recorded in Henan Province during 1971–2020. It is assumed that the annual yield increase is also affected by seed regeneration, chemical fertilizer, and other human factors, and it is showing a linear growth trend, i.e., linear yield (see red line) [29, 44]. The effect of climate on maize yields can be expressed as a weather yield ratio $\eta$ which can be calculated as

$$\eta_i = -\frac{YR_i - YL_i}{YL_i} \tag{15}$$

where $YR$ represents the raw yield and $YL$ represents the linear prediction. The larger the $\eta$,

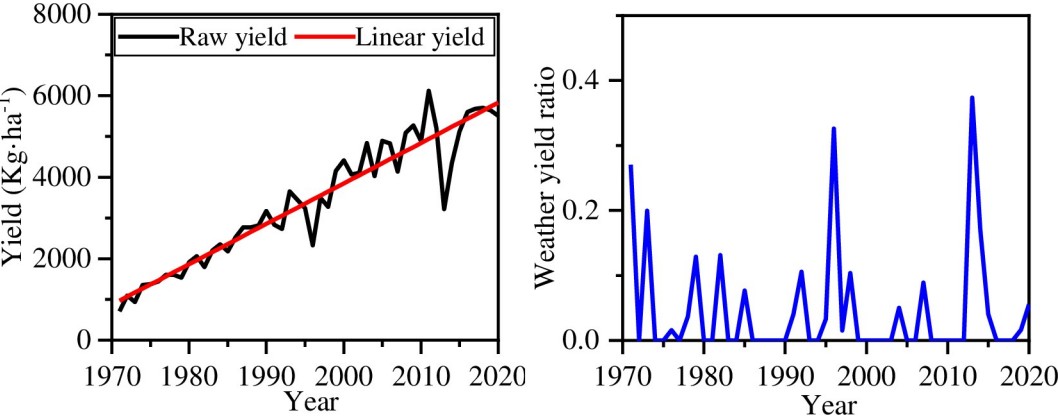

**Fig 1.** Maize yield in Henan Province over 50 years a.) raw yield vs. linear yield, b.) weather yield ratio.

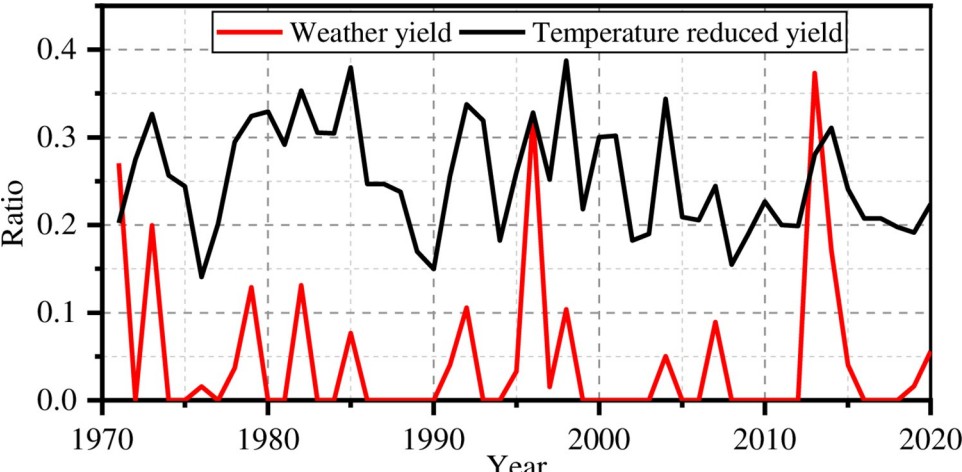

**Fig 2. Temperature reduced yield index and the weather yield ratio.**

the more the maize was affected by unfavorable weather. The weather yield ratio is set to zero when $\eta < 0$, and it is an index between 0 and 1.

The four individual reduced yield indices are calculated using the four climatic suitability models. Based on the meteorological data of Henan Province from 1971 to 2020, the annual temperature reduced yield index $D_t$ of maize can be calculated using Eqs 1–3 and historical meteorological data for Henan Province from 1971 to 2020. The relationship between the temperature reduced yield index $D_t$ and the weather yield ratio is shown in Fig 2. There is clearly a strong correlation between $D_t$ and the weather yield ratio. The correlation coefficient is 0.46. That shows that the temperature suitability model can correctly simulate the trends in the weather yield ratio, though the simulated index is higher. The possible reason is that the suitable temperature range of maize adopted in the current study is rather small, and maize has a better ability to resist temperature changes.

The relationship between the precipitation reduced yield index $D_p$ and the weather yield ratio is shown in Fig 3. The correlation coefficient there is 0.39. The index is again a little higher than the weather yield ratio, which may be due to the drought tolerance of maize.

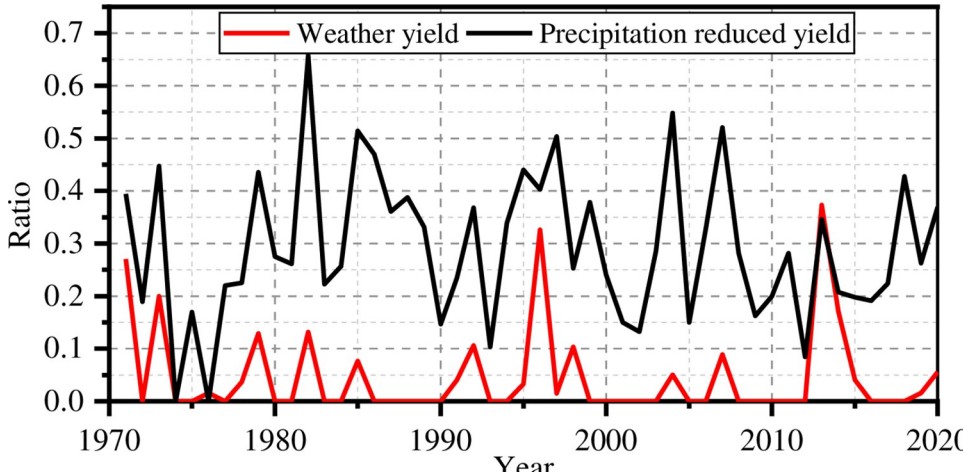

**Fig 3. Precipitation yield reduction index and weather yield ratio.**

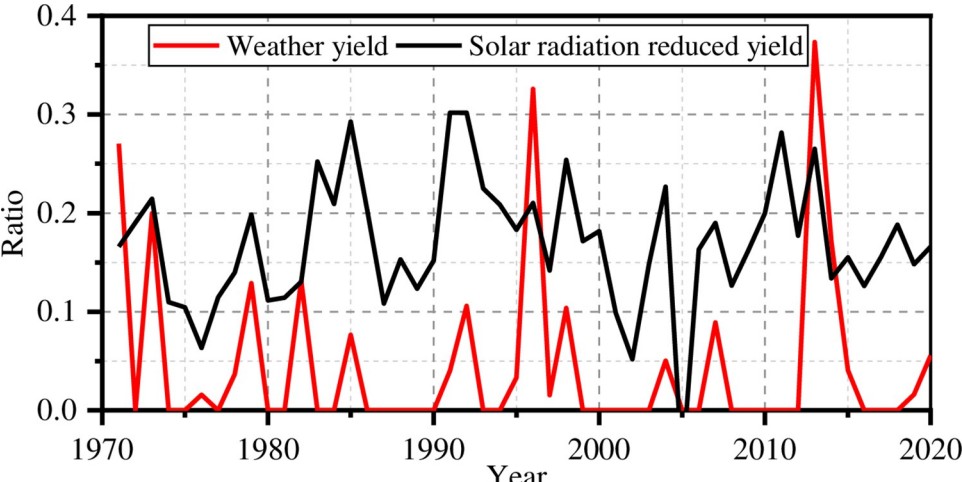

**Fig 4. Solar radiation reduced yield index and the weather yield ratio.**

Fig 4 shows comparable plots for solar radiation reduced yield index $D_s$ and weather yield ratio. The correlation coefficient between solar radiation reduced yield index and weather yield ratio is 0.38. The precipitation reduced yield index best simulates the peak values of the weather yield ratio, but their trends show a certain correlation.

The relationship between the wind reduced yield index $D_w$ and weather yield ratio is shown in Fig 5. The correlation coefficient is 0.24. Their correlation is poor because the wind is only critical during the flowering and pollination period. The effect of wind on the whole growth cycle is limited.

The order of importance of the four factors is temperature, precipitation, solar radiation, and wind speed. Temperature and precipitation dominate over the whole growth cycle of maize.

Two different reduced yield models can be used to simulate the effect of climate change on maize yield. Using the comprehensive reduced yield (CRY) model proposed in the current study, the reduced yield model can be calculated using Eq 9. The CRY model is employed to

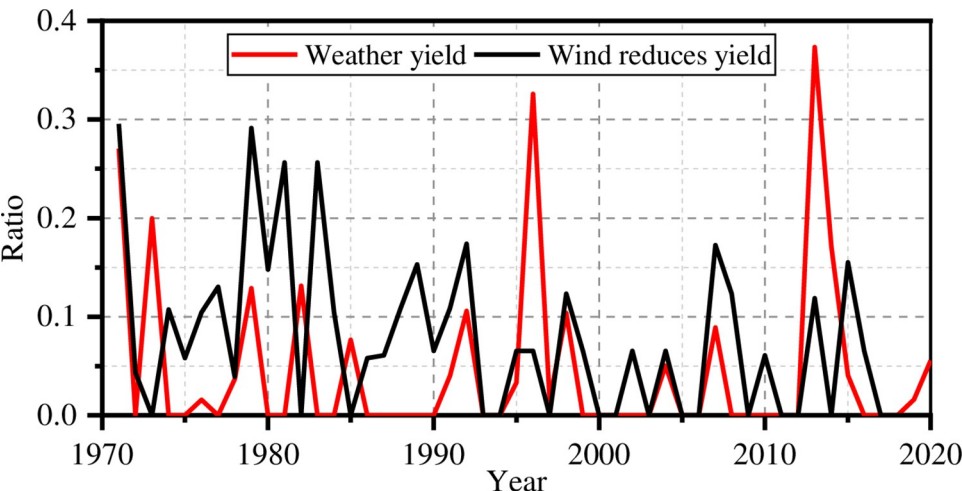

**Fig 5. Wind reduced yield ratio and the weather yield ratio.**

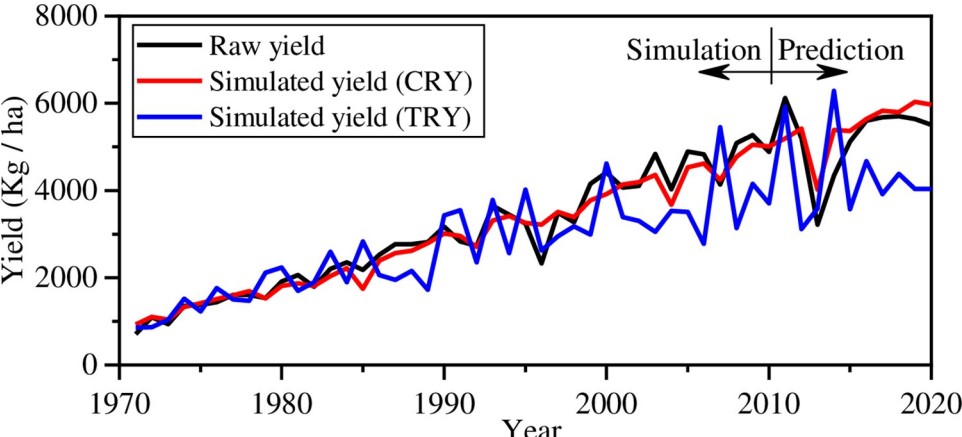

**Fig 6. Raw yield vs. simulated yield.**

simulate the field yield from 1971–2020 affected by climate change after reverse analysis training uses yield from 1971–2010. The traditional reduced yield (TRY) model in D can be calculated as $D = \sqrt[4]{D_t D_r D_s D_w}$ proposed in previous literature [28–30, 34, 40]. The yields predicted by the two reduced yield models are shown in Fig 6.

The correlation coefficient between raw yield and simulated yield obtained by the CRY model is 0.75 which is larger than that obtained by the TRY model (0.37). It illustrates that the comprehensive reduced yield model can well simulate the field yield affected by climate change. The correlation coefficient between raw yield and the CRY predictions is higher than that between any single weather factor index and the raw yield. The CRY model simulates the raw yields from 2010 to 2020 well because it considers the correlations among various weather factors based on their weighted average and reverses analysis.

To study the sensitivities of maize yield to climate change in Henan Province, a direct differentiation method was used to study the sensitivity of the comprehensive climate suitability (CCS) model based on the direct differentiation method. The sensitivities of the weather yield ratio obtained by Eq 13 are shown in Fig 7. The sensitivities to temperature and precipitation are great, and the change is obvious. It indicates that maize yield is sensitive to changes in

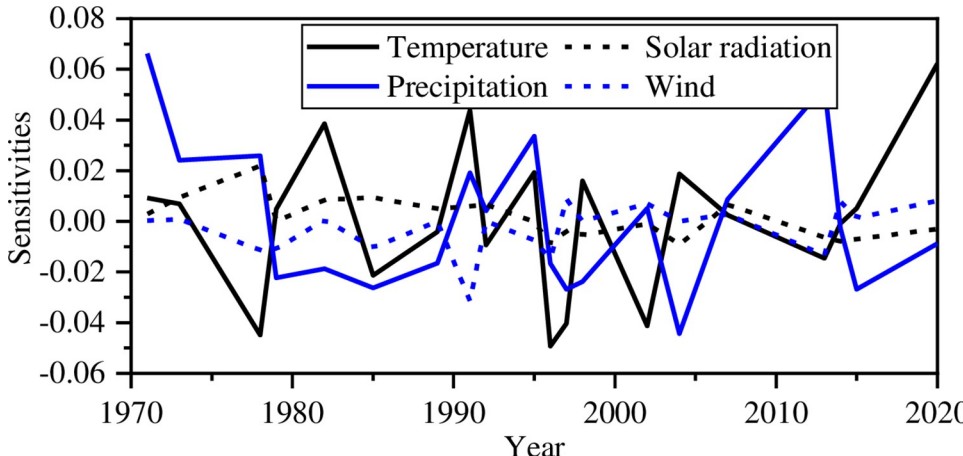

**Fig 7. The sensitivities of the weather yield ratio to weather factors in different years.**

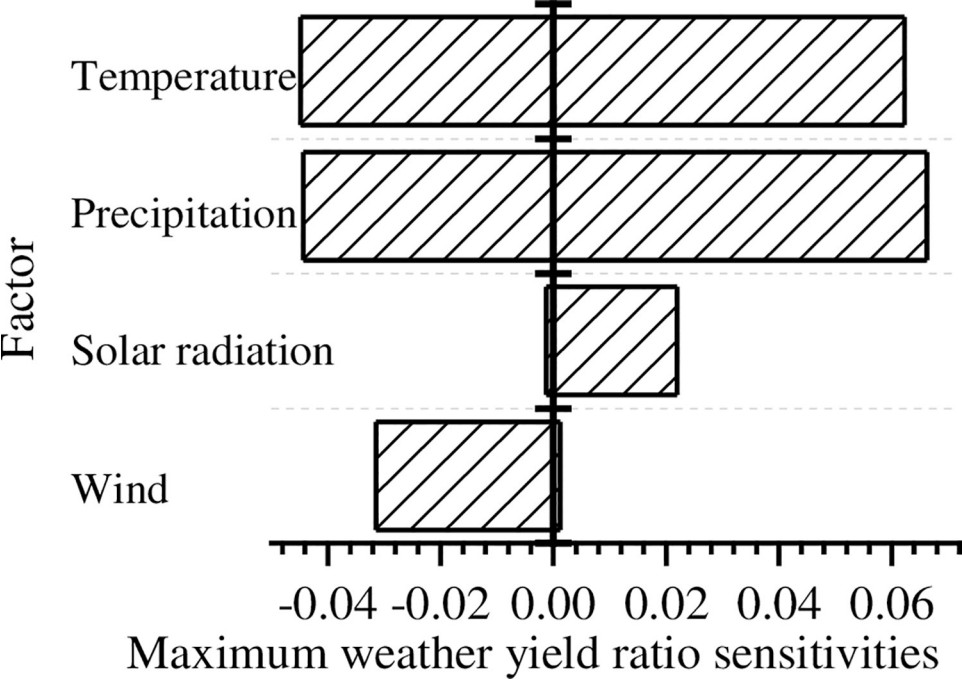

**Fig 8. Maximum weather yield ratio sensitivity to the four factors.**

temperature and precipitation. However, the effect of a one-unit change in solar radiation or wind on maize yield is relatively limited.

The maximum values of the sensitivities over the past 50 years are shown in Fig 8. It shows that temperature and precipitation have the greatest impact on the weather yield ratio. The fluctuation of temperature and precipitation may lead to the decrease of maize yield in different years (through frostbite, withering, drought, waterlogging, and other disasters). Solar radiation and wind are not often important, though in general an increase in light hours will increase yields and higher wind speeds will decrease them [28]. The conclusion of the sensitivity analysis is consistent with actual crop growth characteristics.

The maize comprehensive suitability model proposed in this article can simulate historical data and predict maize yields affected by climate change. The sensitivity analysis results obtained by the sensitivity algorithm of the CCS model illustrate that temperature and precipitation are the dominant weather variables affecting yields. However, too high temperature and precipitation will also have a negative impact on the yield, e.g., through withering, drought, waterlogging and other disasters. Appropriate reduction of wind speed and increase of light intensity is conducive to the increase of yield. The above conclusions have some regional particularities, and they can only be used as a reference for maize yield analysis in Henan Province.

## Conclusions

A comprehensive climate suitability model has been developed for maize in Henan Province based on its temperature, precipitation, solar radiation, and wind requirements in different growth periods. Long-term observation of agrometeorological data and recorded field yields well verify that the maize comprehensive suitability model can better simulate observed historical maize yields affected by climate change. The maize predicted forecast obtained by the CCS

model can match the recorded field yield very well from 1971–2020. Applying the sensitivity algorithm based on the direct differentiation method to the CCS model, the sensitivity analysis shows that temperature and precipitation are the dominant climate factors affecting yield changes, while less sunlight and stronger winds will adversely affect crop yields to some extent. The maize comprehensive suitability model and its sensitivity algorithm can provide an efficient analysis tool for predicting yield reductions due to climate change.

## Supporting information

**S1 Data.**
(CSV)

**S2 Data.**
(CSV)

## Author Contributions

**Conceptualization:** Ning Zhang.

**Data curation:** Ning Zhang, Yizhong Qu, Jiang Jiang.

**Formal analysis:** Ning Zhang.

**Methodology:** Ning Zhang.

**Resources:** Ning Zhang, Yizhong Qu, Jiang Jiang.

**Supervision:** Zhizhong Song.

**Validation:** Zhizhong Song, Yahui Chen.

**Visualization:** Ning Zhang, Zhizhong Song, Yahui Chen.

**Writing – original draft:** Ning Zhang, Yahui Chen.

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
