## [Decision Letter · Decision Letter 0]

6 Dec 2021

PONE-D-21-35009Responses and sensitivities of maize phenology to climate change from 1971 to 2020 in Henan Province, ChinaPLOS ONE

Dear Dr. Zhang,

Thank you for submitting your manuscript to PLOS ONE. After careful consideration, we feel that it has merit but does not fully meet PLOS ONE’s publication criteria as it currently stands. Therefore, we invite you to submit a revised version of the manuscript that addresses the points raised during the review process.

We look forward to receiving your revised manuscript.

Kind regards,

Vassilis G. Aschonitis

Academic Editor

PLOS ONE

Journal Requirements:

("The authors gratefully acknowledge financial support from China’s National Key Research and Development Program under grant 2016YFC0701106 and from China’s National Science Foundation through grants 51978591 and 51578473.")

("The authors gratefully acknowledge financial support from China’s National Key Research and Development Program under grant 2016YFC0701106 and from China’s National Science Foundation through grants 51978591 and 51578473.")

Reviewers' comments:

Reviewer's Responses to Questions

**Comments to the Author**

1. Is the manuscript technically sound, and do the data support the conclusions?

Reviewer #1: Partly

Reviewer #2: Yes

Reviewer #3: Yes

2. Has the statistical analysis been performed appropriately and rigorously? 

Reviewer #1: Yes

Reviewer #2: Yes

Reviewer #3: Yes

3. Have the authors made all data underlying the findings in their manuscript fully available?

Reviewer #1: No

Reviewer #2: Yes

Reviewer #3: Yes

4. Is the manuscript presented in an intelligible fashion and written in standard English?

Reviewer #1: No

Reviewer #2: Yes

Reviewer #3: Yes

5. Review Comments to the Author

Reviewer #1: Recommendation: Reject

The survey reports an interesting topic that points out the effect of climate change on maize phenology and its sensitivity. The manuscript presents a certain gap in standard English. A large amount of reliable references lacks to statements making the manuscript with low basis on previous study (that are largely found). The manuscript didn’t show any effect of climate change on detailed phenological stages. It only showed the effect of climate change on yield and on all phenological stages. It is a huge gap that lead me to take my decision concerning the manuscript. Additionally, authors mentioned several times the term “weather” which is not the case of studying meteorological parameters over a 50 years-period (being climate rather than weather). Moreover, results weren’t approximately discussed nor compared to previous studies, which was also one of the major gaps of the manuscript. Authors mentioned that all the data were present in the manuscript and its supporting files; while it is not the case as the data on which they relied to present the predictions and all presented models isn’t available. I think that the manuscript doesn’t fully merit to be published in PLOS One; however, I ask authors to take into consideration the following comments which will help them to improve their manuscript.

1. Page 1, line 12: Kindly adjust as follow: “ultimately its yield”.

2. Page 1, lines 13–14: Kindly replace “in different growth periods of maize” by “in different phenological stages of maize”.

3. Page 1, line 14: Kindly replace “historical weather” by “climate” as the climate is a state of weather at a very long time starting 30 years and more.

4. Page 1, line 15: Kindly adjust the sentence as follow: “The comprehensive suitability model showed capability of correctly…”

5. Page 1, lines 15–18: “The comprehensive… data”: The sentence is quite long; accordingly, kindly reformulate it to be more concise.

6. Page 1, lines 18–19: “The correlation… are analyzed”: Kindly mention the type of correlation and its strength in the abstract part. You cannot mention it as an analyzed factor while the objective is to mention what did you find in the abstract’s results mention.

7. Page 1, lines 19–20: “The sensitivities… method”: Same recommendation as mentioned in the previous comment.

8. Page 3, lines 37–42: “Many… quality”: This whole section lacks reliable sources (references); accordingly, kindly provide them.

9. Page 3, line 42: Kindly adjust as follow: “Gu et al. [13,14]”.

10. Page 3, line 43: “for growing particular crops”: Kindly mention these crops.

11. Page 3, line 43: Kindly adjust the sentence as follow: “Gonzalez et al. [15] established a model…”

12. Page 3, line 46: Kindly remove reference 16 as it has no relation with the mentioned statement.

13. Page 3, line 46: Kindly adjust the sentence as follow: “Lamptey [17] used…”

14. Page 4, lines 48–49: Kindly adjust as follow: “Wang et al. [18]”.

15. Page 4, line 51: Kindly remove “[18]”, adjust as follow: “Robin [19] and provide the corresponding reliable source “19” in the references list as the one present is different than mentioned in the text.

16. Page 4, line 55: Kindly remove “but”.

17. Page 4, lines 58–59: Kindly adjust the sentence as follow: “The model is mechanistic and takes into account thermal effects, insolation, and different varieties [20–22].”

18. Page 4, line 59: Kindly adjust the sentence as follow: “Zhao et al. [23] used fuzzy…”

19. Page 4, line 61: Kindly remove “[23]”.

20. Page 4, line 62: Kindly adjust as follow: “Yao et al. [24]”.

21. Page 4, line 65: Kindly remove “[24]”.

22. Page 4, lines 65–66: Kindly adjust as follow: “systematic research on…”

23. Page 4, line 66: Kindly remove “That was the object of this study.”

24. Pages 4–5, lines 69–71: “The model… corn”: These statements are related to the Materials and methods section as they describe the approaches take on. Accordingly, kindly remove them.

25. Page 5: Kindly add the title: “2. Materials and methods”.

26. Page 5, line 74: Kindly adjust the title numbering as: “2.1. A comprehensive climate suitability (CCS) model for maize”.

27. Page 5, lines 75–76: “Kindly adjust the sentence as follow: “its whole growth period can be divided…”

28. Page 5, line 79: Kindly adjust “considers” and remove “should”.

29. Page 5, lines 82–83: Kindly adjust the sentence as follow: “Therefore, the independent suitability models need to be first established for the four factors.”

30. Page 5, line 84: Kindly adjust the title numbering as: “2.2. Temperature suitability model”.

31. Page 6, line 103: Kindly adjust the title numbering as: “2.3. Precipitation suitability model”.

32. Page 6, lines 104–106: “Winter… accumulation”: These statements lack reliable sources (references); accordingly, kindly provide them.

33. Page 7, line 108: Kindly adjust as follow: “Wei et al. [26] suggested…”

34. Page 7, line 108: The sentence is incomplete; accordingly, kindly reformulate it.

35. Page 7, lines 108–109: Kindly remove “In this paper” and adjust the sentence as follow: “the precipitation suitability was calculated using the following equation”.

36. Page 7, lines 114–115: “That generates… Table 2”: The sentence is badly written in standard English; accordingly, kindly reformulate it.

37. Page 7, line 121: Kindly adjust the title numbering as: “2.4. Sunshine suitability model”.

38. Pages 7–8, lines 122–126: “Maize… maize”: These statements lack reliable sources; accordingly, kindly provide them.

39. Page 8, line 124: Kindly remove “every day”.

40. Page 8, line 125: Kindly adjust as follow: “inadequate” and “slows down”.

41. Page 8, lines 127–129: “In a certain… suitable”: These statement lack reliable sources; accordingly, kindly provide them.

42. Page 8, lines 129–130: Kindly adjust as follow: “Li et al. [27] suggested…”

43. Page 8, line 138: Kindly provide the source of Table 3 parameters details.

44. Page 9, line 143: Kindly adjust the title numbering as: “2.5. Wind suitability model”.

45. Page 9, lines 144–148: “Wind… variety”: These statements lack reliable sources (references); accordingly, kindly provide them.

46. Page 9, lines 148–149: Kindly adjust this sentence as follow: “In the current study, focus was made on flowering…”

47. Page 9, lines 149–151: “During that… yield”: These statements lack reliable sources (references); accordingly, kindly provide them.

48. Page 9, lines 154–155: “which are taken as 6.0m/s and 10m/s in this discussion”: Kindly remove “in this discussion” and mention the source you relied on to state these numbers as suitable to adopt.

49. Page 9, line 157: “where nw is the number of days in the flowering period—15 days in this discussion”: Kindly remove “in this discussion” and mention the source you relied on to state this number as suitable to adopt.

50. Page 9, line 158: Kindly adjust the title numbering as: “2.6. A comprehensive reduced yield model”.

51. Page 10, lines 159–163: “The ideal… correlation”: These statements lack reliable sources (references); accordingly, kindly provide them.

52. Page 10, line 173: Kindly adjust the title numbering as: “2.7. Sensitivities of comprehensive reduced yield model”.

53. Page 10, lines 174–178: “The climatic… one unit”: These statements lack reliable sources (references); accordingly, kindly provide them.

54. Page 10, line 178: Kindly replace “In this paper” by “In the current study”.

55. Page 11, line 191: Kindly replace this title by “3. Results and discussion”.

56. Page 11, lines 194–195: “It is assumed… factors”: This statement lacks a reliable source (reference); accordingly, kindly provide it.

57. Page 11, line 199: Kindly adjust as follow: “YL”.

58. Page 11, line 200: You’re discussing a climate change over 50 years, you cannot say “weather” !!

59. Page 12, line 206: Kindly remove “firstly”.

60. Page 12, line 207: “Based on the meteorological data of Henan Province from 1971 to 2020”: We are not able to observe the detailed data in these years; accordingly, kindly provide them in a supplementary file”.

61. Page 12, line 209: Kindly adjust as follow: “The relationship between…”

62. Page 12, line 214: Kindly replace “set in this paper” by “adopted in the current study”.

63. Page 14, line 229: Kindly adjust as follow: “The relationship between…”

64. Page 14, lines 230–232: “That is not… 15 days”: The sentence is badly written in standard English; accordingly, kindly reformulate it.

65. Page 14, line 236: Kindly remove “So”.

66. Page 14, line 239: Kindly replace “in this paper” by “in the current study”.

67. Page 15, line 240: Kindly adjust as follow: “to simulate”.

68. Page 15, line 243: Kindly adjust as follow: “in previous literature”.

69. Page 16, line 264: Kindly adjust as follow: “The fluctuation of temperature and precipitation”.

70. Page 16, lines 264–267: “The decrease… decrease them”: These statements lack reliable sources (references); accordingly, kindly provide them.

71. Page 17, line 271: Kindly remove “In summary”.

72. Page 17, lines 271–279: This whole section is a summary of the current’s study findings. Kindly remove it.

Reviewer #2: "Responses and sensitivities of maize phenology to climate change from 1971 to 2020 in Henan Province, China" is well written. It is about the climate change effects on maize under particular conditions of China.

Reviewer #3: Dear author

In a general review of your paper, I found that the computational and mathematical aspects of this paper are predominant, and such writing certainly demonstrates the authors' mastery of the concepts of simulation and the use of performance prediction models using real farm multi-year data.

Therefore, considering the importance of predicting the change of yield of important crops such as corn in the face of climate change, I consider this article to be very important and practical. However, it is necessary to use specialized words that are generally accepted by experts in this field in expressing biological topics. I have suggested some tips in this regard.

In addition, I found the method of referring to the research of others unconventional in the text of your article. However, to address this shortcoming, I have written some suggestions for you that I hope you will find useful.

6. PLOS authors have the option to publish the peer review history of their article (what does this mean?). If published, this will include your full peer review and any attached files.

Reviewer #1: No

Reviewer #2: No

Reviewer #3: **Yes: **Majid AghaAlikhani(Ph.D.)

Professor

Weed-Crop Ecophysiology

Medicinal and Aromatic Plants Agrotechnology

Department of Agrotechnology

Faculty of Agriculture

Tarbiat Modares University

PO Box 14115-336

Tehran, Iran

Phone:+98 21 48292099

---

## [Author Response · Author response to Decision Letter 0]

14 Dec 2021

Journal: PLOS ONE

Title: Responses and sensitivities of maize phenology to climate change from 1971 to 2020 in Henan Province, China

Manuscript ID: PONE-D-21-35009

Dear Prof. Vassilis G. Aschonitis,

Thank you very much for your letter that enclosed the reviewers’ comments for our manuscript. We have studied the comments carefully and tried our best to improve the manuscript. We would like to submit the revised manuscript with all revisions marked in blue color. The following are our responses to the reviewers’ comments. Some essential contents have been added in the revised manuscript, however, these changes will not influence the conclusions. 

We wish to take this opportunity to thank you and all the reviewers for your consideration and instructive comments. Look forward to hearing from you! 

Sincerely,

Ning Zhang

Doctor

Faculty of Forestry, The University of British Columbia, 

Vancouver, British Columbia, Canada 

 

Additional requirements

Response: Thanks for your reminder. The manuscript has been revised to meet PLOS ONE's style requirement.

Response: The Funding Information has been revised.

("The authors gratefully acknowledge financial support from China’s National Key Research and Development Program under grant 2016YFC0701106 and from China’s National Science Foundation through grants 51978591 and 51578473.")

("The authors gratefully acknowledge financial support from China’s National Key Research and Development Program under grant 2016YFC0701106 and from China’s National Science Foundation through grants 51978591 and 51578473.")

Response: The funding-related text has been removed from the manuscript. 

In the Funding Statement,

“The authors gratefully acknowledge financial support from China’s National Science Foundation through grants 51978591 and 51578473.”

Response:

Data Availability Statement

All data, models, or codes that support the findings of this study are available. The functions of the model are explained in detail in the manuscript. The data of weather and yield are attached as supporting information.

 

To Reviewer 1

Reviewer #1: The survey reports an interesting topic that points out the effect of climate change on maize phenology and its sensitivity. The manuscript presents a certain gap in standard English. A large amount of reliable references lacks to statements making the manuscript with low basis on previous study (that are largely found). The manuscript didn’t show any effect of climate change on detailed phenological stages. It only showed the effect of climate change on yield and on all phenological stages. It is a huge gap that lead me to take my decision concerning the manuscript. Additionally, authors mentioned several times the term “weather” which is not the case of studying meteorological parameters over a 50 years-period (being climate rather than weather). Moreover, results weren’t approximately discussed nor compared to previous studies, which was also one of the major gaps of the manuscript. Authors mentioned that all the data were present in the manuscript and its supporting files; while it is not the case as the data on which they relied to present the predictions and all presented models isn’t available. I think that the manuscript doesn’t fully merit to be published in PLOS One; however, I ask authors to take into consideration the following comments which will help them to improve their manuscript.

Response: 

We have studied the comments carefully and tried our best to improve the manuscript. These suggestions are very helpful for us to improve our manuscript. We wish to take this opportunity to thank you for your consideration and instructive comments. The statement about “climate” is incorrect, and it has been replaced by “weather”. For Data Availability Statement, all data, models, or code that support the findings of this study are available from the corresponding author upon reasonable request. We would like to submit the revised manuscript with all revisions marked in blue color. 

1. Page 1, line 12: Kindly adjust as follow: “ultimately its yield”.

Response: Thanks for your comment. It has been revised in the manuscript.

2. Page 1, lines 13–14: Kindly replace “in different growth periods of maize” by “in different phenological stages of maize”.

Response: It has been revised in the manuscript. Thanks.

3. Page 1, line 14: Kindly replace “historical weather” by “climate” as the climate is a state of weather at a very long time starting 30 years and more.

4. Page 1, line 15: Kindly adjust the sentence as follow: “The comprehensive suitability model showed capability of correctly…”

5. Page 1, lines 15–18: “The comprehensive… data”: The sentence is quite long; accordingly, kindly reformulate it to be more concise.

Response: The author agrees with the revises of questions 3-5.

6. Page 1, lines 18–19: “The correlation… are analyzed”: Kindly mention the type of correlation and its strength in the abstract part. You cannot mention it as an analyzed factor while the objective is to mention what did you find in the abstract’s results mention.

Response: It has been replaced by “The results of correlation showed that the yields are more closely related to multi-weather factors, temperature and precipitation than to solar radiation and wind. “

7. Page 1, lines 19–20: “The sensitivities… method”: Same recommendation as mentioned in the previous comment.

Response: It has been replaced by “The sensitivity analysis illustrates that temperature and precipitation are the dominant weather factors affecting yield changes based on a direct differentiation method. ”

8. Page 3, lines 37–42: “Many… quality”: This whole section lacks reliable sources (references); accordingly, kindly provide them.

Response: Five literature have been added for reference, see Ref.29-33.

9. Page 3, line 42: Kindly adjust as follow: “Gu et al. [13,14]”.

Response: It has been revised in the manuscript.

10. Page 3, line 43: “for growing particular crops”: Kindly mention these crops.

Response: It has been revised in the manuscript, such as maize and wheat.

11. Page 3, line 43: Kindly adjust the sentence as follow: “Gonzalez et al. [15] established a model…”

Response: They have been revised in the manuscript, including Questions 12-72.

12. Page 3, line 46: Kindly remove reference 16 as it has no relation with the mentioned statement.

13. Page 3, line 46: Kindly adjust the sentence as follow: “Lamptey [17] used…”

14. Page 4, lines 48–49: Kindly adjust as follow: “Wang et al. [18]”.

15. Page 4, line 51: Kindly remove “[18]”, adjust as follow: “Robin [19] and provide the corresponding reliable source “19” in the references list as the one present is different than mentioned in the text.

16. Page 4, line 55: Kindly remove “but”.

17. Page 4, lines 58–59: Kindly adjust the sentence as follow: “The model is mechanistic and takes into account thermal effects, insolation, and different varieties [20–22].”

18. Page 4, line 59: Kindly adjust the sentence as follow: “Zhao et al. [23] used fuzzy…”

19. Page 4, line 61: Kindly remove “[23]”.

20. Page 4, line 62: Kindly adjust as follow: “Yao et al. [24]”.

21. Page 4, line 65: Kindly remove “[24]”.

22. Page 4, lines 65–66: Kindly adjust as follow: “systematic research on…”

23. Page 4, line 66: Kindly remove “That was the object of this study.”

24. Pages 4–5, lines 69–71: “The model… corn”: These statements are related to the Materials and methods section as they describe the approaches take on. Accordingly, kindly remove them.

25. Page 5: Kindly add the title: “2. Materials and methods”.

26. Page 5, line 74: Kindly adjust the title numbering as: “2.1. A comprehensive climate suitability (CCS) model for maize”.

27. Page 5, lines 75–76: “Kindly adjust the sentence as follow: “its whole growth period can be divided…”

28. Page 5, line 79: Kindly adjust “considers” and remove “should”.

29. Page 5, lines 82–83: Kindly adjust the sentence as follow: “Therefore, the independent suitability models need to be first established for the four factors.”

30. Page 5, line 84: Kindly adjust the title numbering as: “2.2. Temperature suitability model”.

31. Page 6, line 103: Kindly adjust the title numbering as: “2.3. Precipitation suitability model”.

32. Page 6, lines 104–106: “Winter… accumulation”: These statements lack reliable sources (references); accordingly, kindly provide them.

33. Page 7, line 108: Kindly adjust as follow: “Wei et al. [26] suggested…”

34. Page 7, line 108: The sentence is incomplete; accordingly, kindly reformulate it.

35. Page 7, lines 108–109: Kindly remove “In this paper” and adjust the sentence as follow: “the precipitation suitability was calculated using the following equation”.

36. Page 7, lines 114–115: “That generates… Table 2”: The sentence is badly written in standard English; accordingly, kindly reformulate it.

37. Page 7, line 121: Kindly adjust the title numbering as: “2.4. Sunshine suitability model”.

38. Pages 7–8, lines 122–126: “Maize… maize”: These statements lack reliable sources; accordingly, kindly provide them.

39. Page 8, line 124: Kindly remove “every day”.

40. Page 8, line 125: Kindly adjust as follow: “inadequate” and “slows down”.

41. Page 8, lines 127–129: “In a certain… suitable”: These statement lack reliable sources; accordingly, kindly provide them.

42. Page 8, lines 129–130: Kindly adjust as follow: “Li et al. [27] suggested…”

43. Page 8, line 138: Kindly provide the source of Table 3 parameters details.

44. Page 9, line 143: Kindly adjust the title numbering as: “2.5. Wind suitability model”.

45. Page 9, lines 144–148: “Wind… variety”: These statements lack reliable sources (references); accordingly, kindly provide them.

46. Page 9, lines 148–149: Kindly adjust this sentence as follow: “In the current study, focus was made on flowering…”

47. Page 9, lines 149–151: “During that… yield”: These statements lack reliable sources (references); accordingly, kindly provide them.

48. Page 9, lines 154–155: “which are taken as 6.0m/s and 10m/s in this discussion”: Kindly remove “in this discussion” and mention the source you relied on to state these numbers as suitable to adopt.

49. Page 9, line 157: “where nw is the number of days in the flowering period—15 days in this discussion”: Kindly remove “in this discussion” and mention the source you relied on to state this number as suitable to adopt.

50. Page 9, line 158: Kindly adjust the title numbering as: “2.6. A comprehensive reduced yield model”.

51. Page 10, lines 159–163: “The ideal… correlation”: These statements lack reliable sources (references); accordingly, kindly provide them.

52. Page 10, line 173: Kindly adjust the title numbering as: “2.7. Sensitivities of comprehensive reduced yield model”.

53. Page 10, lines 174–178: “The climatic… one unit”: These statements lack reliable sources (references); accordingly, kindly provide them.

54. Page 10, line 178: Kindly replace “In this paper” by “In the current study”.

55. Page 11, line 191: Kindly replace this title by “3. Results and discussion”.

56. Page 11, lines 194–195: “It is assumed… factors”: This statement lacks a reliable source (reference); accordingly, kindly provide it.

57. Page 11, line 199: Kindly adjust as follow: “YL”.

58. Page 11, line 200: You’re discussing a climate change over 50 years, you cannot say “weather” !!

59. Page 12, line 206: Kindly remove “firstly”.

60. Page 12, line 207: “Based on the meteorological data of Henan Province from 1971 to 2020”: We are not able to observe the detailed data in these years; accordingly, kindly provide them in a supplementary file”.

61. Page 12, line 209: Kindly adjust as follow: “The relationship between…”

62. Page 12, line 214: Kindly replace “set in this paper” by “adopted in the current study”.

63. Page 14, line 229: Kindly adjust as follow: “The relationship between…”

64. Page 14, lines 230–232: “That is not… 15 days”: The sentence is badly written in standard English; accordingly, kindly reformulate it.

65. Page 14, line 236: Kindly remove “So”.

66. Page 14, line 239: Kindly replace “in this paper” by “in the current study”.

67. Page 15, line 240: Kindly adjust as follow: “to simulate”.

68. Page 15, line 243: Kindly adjust as follow: “in previous literature”.

69. Page 16, line 264: Kindly adjust as follow: “The fluctuation of temperature and precipitation”.

70. Page 16, lines 264–267: “The decrease… decrease them”: These statements lack reliable sources (references); accordingly, kindly provide them.

71. Page 17, line 271: Kindly remove “In summary”.

72. Page 17, lines 271–279: This whole section is a summary of the current’s study findings. Kindly remove it.

Response: We would like to express our grateful thanks to the reviewer for the above suggestions. We have studied the comments carefully and tried our best to improve the manuscript. These suggestions are very helpful for us to improve our manuscript. 

 

To Reviewer 2

Reviewer #2: "Responses and sensitivities of maize phenology to climate change from 1971 to 2020 in Henan Province, China" is well written. It is about the climate change effects on maize under particular conditions of China.

Response: 

We would like to express our grateful thanks to the reviewer for his/her valuable comments. All the doubts and remarks have been answered carefully. The numbers for sections, figures, tables, and pages refer to the revised version of the manuscript unless otherwise noted. The modifications are highlighted in blue in the revised manuscript.

 

To Reviewer 3

Reviewer #3: Dear author

In a general review of your paper, I found that the computational and mathematical aspects of this paper are predominant, and such writing certainly demonstrates the authors' mastery of the concepts of simulation and the use of performance prediction models using real farm multi-year data.

Therefore, considering the importance of predicting the change of yield of important crops such as corn in the face of climate change, I consider this article to be very important and practical. However, it is necessary to use specialized words that are generally accepted by experts in this field in expressing biological topics. I have suggested some tips in this regard.

In addition, I found the method of referring to the research of others unconventional in the text of your article. However, to address this shortcoming, I have written some suggestions for you that I hope you will find useful.

Response: 

We have studied the comments carefully and tried our best to improve the manuscript. These suggestions are very helpful for us to improve our manuscript. We wish to take this opportunity to thank you for your consideration and instructive comments. We would like to submit the revised manuscript with all revisions marked in blue color. 

ABSTARCT

- The final sentence of the abstract section should inform the final and practical result of the research. Please rewrite this section.

Response: Thanks for your comment. The final and practical result of the research has been added in the revised abstract.

- Line 13: Replace "sunshine": with the "solar radiation". The later one is more professional word to talk about crop yield and its climatic requirements.

Response: It has been replaced.

- Line 18: Correct predicated…. to ……. Predicted.

Response: The typos have been revised.

KEYWORDS

- As article keywords: It is recommended to use words that are not used in the title.

 Response: The KEYWORDS has been revised. Thanks for your comment.

INTRODUCTION

- Lines 35: Replace appropriate seeds to "appropriate genotypes". 

- Line 39: Replace indexes with " indices"

- Line 40, 56, 61, 64, 79: Replace "sunshine": with the "solar radiation".

 Response: The above three inappropriate words have been revised.

- Line 42: Change, Gu and his colleagues… to " Gu et al. It seems the journal guidelines have preferred the number-based citation format, and you are accustomed to reviewing sources, start with citations to the authors' names. It is recommended that you use passive verbs to avoid this inconsistency. For example, instead of "Gu and his colleagues used the climate model," say " the climate model was used ( ). And at the end of the sentence, bring the relevant reference number.

- Line 43: same correction as Line 42 is needed for Gonzales…. 

- Line 46: same correction as Line 42 is needed for Lamptey…. 

- Line 47: Correct Yields ---------� yield

- Line 48: same correction as Line 42 is needed for Wang's group…..

- Line 54: same correction as Line 42 is needed for Robin ……

- Line 56: Replace " some scholars with " a few researcher"

- Line 56-61: no conventional citation: Agroup led by Zhao …. . Same correction as Line 42 is needed for that.

- Line 62: same correction as Line 42 is needed for Yao …… 

 Response: The method of referring has been revised.

- Line 68: Delete the repeated word "effect". So rewrite the sentence having the shared word "effect" for all parameters. 

Response: It has been replaced by 

“This paper establishes a comprehensive weather suitability model to quantify the effect of temperature, water, solar radiation, and wind in a maize growth cycle based on the maize physiological growth model.”

- Line 72: Instead of pointing directly to the application of this article, which is an unproven claim, say: The practical purpose of this research is to be able to provide a tool for accurately predicting corn yield in the face of climate change.

Response: Thanks for your comments. The application of this article has been revised.

- Line 73: is there difference between maize yield (line 72) and yield of summer maize (line 73)??

Response: There is no difference between maize and summer maize. The “summer maize” has been replaced by “maize” in this manuscript.

- Line 74: you applied several models to simulate the weather and meteorological parameters effects on maize yield during a 50 years duration. So these part and their subtitles should be named as Methids and Materials.

Response: It has been named in the revised manuscript.

- Line 76: Replace the summer maize with "the crop".

Response: It has been replaced. 

- Line 79, 82, 121, 123, 124, 126, 128, 129, 130, 133, 136, 137, 138, 140, 162, 163 and 165: Replace "sunshine": with the "solar radiation".

Response: It has been replaced. 

- Line 109: same correction as Line 42 is needed for Wei and his colleagues ……

Response: It has been revised. 

- Line 122: Rewrite this simple but important sentence using a better and an academic expression. You severely recommended to emphasizing into the carboxylation pathway of the maize (C4) which indicating its high demand of solar radiation. 

Response: Thanks for your comments and advice. They have been revised in Line 137 of the manuscript. 

- Line 124-125: this concept has been repeated in above line. So delete this sentence: Maize need a large amount of sunshine.

Response: It has been deleted. 

- Line 145: correct the unit format to : m. s-1

Response: It has been revised. 

- Line 148: Replace variety ----------� cultivar

- Line 150: Do you mean the wind speed? You mentioned to the wind force!! They are different. If your answer is YES, write the unit for wind speed: m.s-1

Response: YES. I am sorry for that. The wind force has been replaced by wind speed. 

- Line 154: correct the unit format to : m. s-1

Response: The following questions have been revised.

- Line 160: Change elements with: parameters". 

- Line 168: Replace indexes with " indices"

- Line 171: Replace indexes with " indices"

- Line 178: Change the paper with the " article"

- Line 191: Write the main title for this part of the article : "Results and Discussion"

- Line 200 : Correct radio -------- > ratio

- Figurr 1: correct the unit typesetting as : kg. ha-1

- Line 206: Replace indexes with " indices"

- Line 223, 224, 228, 236, 259, 266: 282 and 289 : Replace "sunshine": with the "solar radiation".

- Figure 7 and figure 8: Replace "sunshine": with the "solar radiation".

- Line 271: Replace the paper with the "article" 

Response: We would like to thank you again for the above suggestions. We have studied the comments carefully and tried our best to improve the manuscript. These suggestions are very helpful for me to improve my manuscript.

---

## [Decision Letter · Decision Letter 1]

19 Dec 2021

PONE-D-21-35009R1Responses and sensitivities of maize phenology to weather change from 1971 to 2020 in Henan Province, ChinaPLOS ONE

Dear Dr. Zhang,

Thank you for submitting your manuscript to PLOS ONE. After careful consideration, we feel that it has merit but does not fully meet PLOS ONE’s publication criteria as it currently stands. Therefore, we invite you to submit a revised version of the manuscript that addresses the points raised during the review process.

We look forward to receiving your revised manuscript.

Kind regards,

Vassilis G. Aschonitis

Academic Editor

PLOS ONE

Journal Requirements:

Reviewers' comments:

Reviewer's Responses to Questions

**Comments to the Author**

1. If the authors have adequately addressed your comments raised in a previous round of review and you feel that this manuscript is now acceptable for publication, you may indicate that here to bypass the “Comments to the Author” section, enter your conflict of interest statement in the “Confidential to Editor” section, and submit your "Accept" recommendation.

Reviewer #1: All comments have been addressed

Reviewer #3: All comments have been addressed

2. Is the manuscript technically sound, and do the data support the conclusions?

Reviewer #1: Yes

Reviewer #3: Yes

3. Has the statistical analysis been performed appropriately and rigorously? 

Reviewer #1: Yes

Reviewer #3: Yes

4. Have the authors made all data underlying the findings in their manuscript fully available?

Reviewer #1: Yes

Reviewer #3: Yes

5. Is the manuscript presented in an intelligible fashion and written in standard English?

Reviewer #1: Yes

Reviewer #3: Yes

6. Review Comments to the Author

Reviewer #1: Recommendation: Accept

After careful and deep observation of all the authors corrections, for which I am deeply grateful for taking all my comments and recommendations into consideration, I find that the manuscript has now a merit to be published in “PLOS One” in its current form. Authors provided all the reliable sources that were lacking in literature (Introduction), Materials and methods and Discussion parts, adjusted all the sentences that were unclear or badly written in standard English and provided all the supporting data that they were asked for as additional files. I am deeply grateful for this big improvement in the manuscript, that made me completely change my mind in its concern.

Reviewer #3: I reviewed the authors' responses to my points and comments, as well as their responses to the suggestions of two other respected referees. I hereby confirm that the authors have taken note of all comments made by the reviewers and have made any necessary changes to the text of the revised version of the article. However, I consider it necessary to pay attention to the following two points and make the relevant correction.

- 1- The author did not understand the meaning of reviewer No. 1, who basically rejected the article, and changed the correct phrase "Climate Change" to the incorrect phrase "weather" in manuscript title, abstract and a few sentences of the body. Do not forget the weather gives temperature, radiation, humidity, etc. in a short period of time, such as 24 a day and 7 day a week. Meanwhile whenever we talk about the status of these parameters in the long run, the term climate should be used. Since this article also uses atmospheric structural data for a period of 50 years, it is appropriate to use the term "climate change" like the original version of the article.

- 2- Although in the text of the article, the typographical error about the word "ratio" that was written as "radio" have been, the authors forgot to correct this point in the relevant figures as well. In this regard, in Figures 2, 3, 4 and 5 in its vertical axis, the word "radio" should be modified to the word "ratio". There is also a grammatical error in the text of the legend inside the figures. In Figures 2, 3, 4, and 5, which refer to reduced yield in terms of temperature, rainfall, solar radiation, and wind, respectively, in the legend, the word "reduces" must be corrected to "reduced".

With best wishes and kind Regards

Reviewer # 3

7. PLOS authors have the option to publish the peer review history of their article (what does this mean?). If published, this will include your full peer review and any attached files.

Reviewer #1: No

Reviewer #3: **Yes: **Majid AghaAlikhani (Ph.D.)

Professor

Weed-Crop Ecophysiology

Medicinal and Aromatic Plants Agrotechnology

Department of Agrotechnology

Faculty of Agriculture

Tarbiat Modares University

PO Box 14115-336

Tehran, Iran

Phone:+98 21 48292099

Facsimile: +98 21 48292200

Mail to: maghaalikhani@modares.ac.ir

Alternate e-mail: majid.aghaalikhani@gmail.com

---

## [Author Response · Author response to Decision Letter 1]

19 Dec 2021

Journal: PLOS ONE

Title: Responses and sensitivities of maize phenology to climate change from 1971 to 2020 in Henan Province, China

Manuscript ID: PONE-D-21-35009

Dear Prof. Vassilis G. Aschonitis,

Thank you very much for your letter that enclosed the reviewers’ comments for our manuscript. We have studied the comments carefully and tried our best to improve the manuscript. We would like to submit the revised manuscript with all revisions marked in blue color. The following are our responses to the reviewers’ comments. Some essential contents have been added in the revised manuscript, however, these changes will not influence the conclusions. 

We wish to take this opportunity to thank you and all the reviewers for your consideration and instructive comments. Look forward to hearing from you! 

Sincerely,

Ning Zhang

Doctor

Faculty of Forestry, The University of British Columbia, 

Vancouver, British Columbia, Canada 

 

To Reviewer 1

Reviewer #1: After careful and deep observation of all the authors corrections, for which I am deeply grateful for taking all my comments and recommendations into consideration, I find that the manuscript has now a merit to be published in “PLOS One” in its current form. Authors provided all the reliable sources that were lacking in literature (Introduction), Materials and methods and Discussion parts, adjusted all the sentences that were unclear or badly written in standard English and provided all the supporting data that they were asked for as additional files. I am deeply grateful for this big improvement in the manuscript, that made me completely change my mind in its concern.

Response: 

We have studied the comments carefully and tried our best to improve the manuscript. We are very excited that you have changed your mind. All of your comments are very helpful to improve our manuscript. Thank you again.

Best wishes

 

To Reviewer 3

Reviewer #3: I reviewed the authors' responses to my points and comments, as well as their responses to the suggestions of two other respected referees. I hereby confirm that the authors have taken note of all comments made by the reviewers and have made any necessary changes to the text of the revised version of the article. However, I consider it necessary to pay attention to the following two points and make the relevant correction.

Response: 

We have studied the comments carefully and tried our best to improve the manuscript. These suggestions are very helpful for us to improve our manuscript. We wish to take this opportunity to thank you for your consideration and instructive comments.

- 1- The author did not understand the meaning of reviewer No. 1, who basically rejected the article, and changed the correct phrase "Climate Change" to the incorrect phrase "weather" in manuscript title, abstract and a few sentences of the body. Do not forget the weather gives temperature, radiation, humidity, etc. in a short period of time, such as 24 a day and 7 day a week. Meanwhile whenever we talk about the status of these parameters in the long run, the term climate should be used. Since this article also uses atmospheric structural data for a period of 50 years, it is appropriate to use the term "climate change" like the original version of the article.

Response: 

We agree with your comments on "Climate Change". As you said, we focus on the atmospheric structural data for a period of 50 years. We believe that "Climate Change" is more appropriate. The “weather” has been replaced by "Climate Change" in the revised manuscript.

Thanks very much for your comments.

- 2- Although in the text of the article, the typographical error about the word "ratio" that was written as "radio" have been, the authors forgot to correct this point in the relevant figures as well. In this regard, in Figures 2, 3, 4 and 5 in its vertical axis, the word "radio" should be modified to the word "ratio". There is also a grammatical error in the text of the legend inside the figures. In Figures 2, 3, 4, and 5, which refer to reduced yield in terms of temperature, rainfall, solar radiation, and wind, respectively, in the legend, the word "reduces" must be corrected to "reduced".

Response: 

Thanks for your comment. " radio" is a typographical error. We are very sorry that we missed this error when we revised the manuscript for the first time. They have been revised in the updated manuscript. In addition, "reduces" has been corrected to "reduced"

---

## [Editor Report · Decision Letter 2]

21 Dec 2021

Responses and sensitivities of maize phenology to climate change from 1971 to 2020 in Henan Province, China

PONE-D-21-35009R2

Dear Dr. Zhang,

We’re pleased to inform you that your manuscript has been judged scientifically suitable for publication and will be formally accepted for publication once it meets all outstanding technical requirements.

Kind regards,

Vassilis G. Aschonitis

Academic Editor

PLOS ONE
---

## [Editor Report · Acceptance letter]

12 Jan 2022

PONE-D-21-35009R2 

Responses and sensitivities of maize phenology to climate change from 1971 to 2020 in Henan Province, China 

Dear Dr. Zhang:

I'm pleased to inform you that your manuscript has been deemed suitable for publication in PLOS ONE. Congratulations! Your manuscript is now with our production department. 

Kind regards, 

on behalf of

Dr. Vassilis G. Aschonitis 

Academic Editor

PLOS ONE